# Identifying the Risk Factors Associated with Nursing Home Residents’ Pressure Ulcers Using Machine Learning Methods

**DOI:** 10.3390/ijerph18062954

**Published:** 2021-03-13

**Authors:** Soo-Kyoung Lee, Juh Hyun Shin, Jinhyun Ahn, Ji Yeon Lee, Dong Eun Jang

**Affiliations:** 1College of Nursing, Keimyung University, 1095 Dalgubeol-daero, Dalseo-gu, Daegu 42601, Korea; soo1005@kmu.ac.kr; 2College of Nursing, Ewha Womans University, Science & Ewha Research Institute of Nursing Science, Seoul 120750, Korea; 3Department of Management Information Systems, Jeju National University, Jeju 63243, Korea; jha@jejunu.ac.kr; 4College of Nursing, Catholic University of Pusan, Busan 46252, Korea; jylee@cup.ac.kr; 5School of Nursing, University of Texas at Austin, Austin, TX 78712, USA; dejang@utexas.edu

**Keywords:** pressure ulcers, machine learning, nursing home

## Abstract

Background: Machine learning (ML) can keep improving predictions and generating automated knowledge via data-driven predictors or decisions. Objective: The purpose of this study was to compare different ML methods including random forest, logistics regression, linear support vector machine (SVM), polynomial SVM, radial SVM, and sigmoid SVM in terms of their accuracy, sensitivity, specificity, negative predictor values, and positive predictive values by validating real datasets to predict factors for pressure ulcers (PUs). Methods: We applied representative ML algorithms (random forest, logistic regression, linear SVM, polynomial SVM, radial SVM, and sigmoid SVM) to develop a prediction model (N = 60). Results: The random forest model showed the greatest accuracy (0.814), followed by logistic regression (0.782), polynomial SVM (0.779), radial SVM (0.770), linear SVM (0.767), and sigmoid SVM (0.674). Conclusions: The random forest model showed the greatest accuracy for predicting PUs in nursing homes (NHs). Diverse factors that predict PUs in NHs including NH characteristics and residents’ characteristics were identified according to diverse ML methods. These factors should be considered to decrease PUs in NH residents.

## 1. Introduction

Pressure ulcers (PUs) are localized injuries of the skin and/or underlying tissue over a bony prominence caused by pressure and shear [1]. The prevalence of physical frailty among nursing home (NH) residents is high (from 19% to 75.6%) [2]. NH residents are often considered a vulnerable population, as they face many losses [3] and are even exposed to various kinds of abuse [4,5]. Preventing PUs—especially in NHs—is important because they contribute not only to morbidity and healthcare costs [6] but also affect residents’ quality of life [7,8]. According to a 10-year survey of PU prevalence in a population of 918,621 NH residents in the US, even though the overall PU prevalence in acute-care and rehabilitation settings decreased significantly, the percentage in long-term-care settings varied and was higher than in acute-care settings (8.8% versus 11.3% in 2015) [9]. Furthermore, 1.7% of NH residents had a suspected deep-tissue injury in 2012 [10]. In Korea, although there is no official data on the incidence of PUs in long-term-care settings, the incidence of PUs has been reported to be 1.2–31.3% in a small number of prior studies [11,12,13].

A systematic review of 54 studies identified the major risk factors for PU development among acute and community patient populations including three primary domains of mobility/activity, perfusion, and skin/PU status [14]. In addition, a secondary analysis study of the 2012 Minimum Data Set reported that four elements of Defloor’s model (compressive forces, shearing forces, tissue tolerance for pressure, and tissue tolerance for oxygen) significantly predicted PUs including suspected deep-tissue injuries in NH residents [10]. Predictors in previous studies were mostly identified using a traditional statistical analysis, which has limitations in handling high-variability data, nonlinear variables, and heterogeneous distributions [15]. Therefore, previous studies heavily depended on the researcher’s knowledge and experience for statistical processing or they might have achieved inaccurate results. A more accurate and efficient method is needed to investigate and better understand PU predictors in long-term-care settings.

Due to advanced technologies and informatics, a new computer-science study method, machine learning (ML), has been developed to overcome the limitations of traditional regression analysis. ML uses statistics, optimization methods, and artificial intelligence to obtain statistical algorithms from an available dataset [15]. Because ML can keep improving predictions and generating automated knowledge via data-driven predictors or decisions, ML is useful for distinguishing relevant dependent variables [16] and supporting clinical decisions [15]. Recently, ML started gaining attention, and researchers from health, medicine, and nursing fields utilize ML because of those advantages. For example, ML has been used to investigate mortality predictors [15], disease prognosis and prediction [17,18,19], emergency department triage prediction [20], and fall prediction [21]. In addition, previous studies using ML concluded that ML showed a superior performance regarding hospital-related outcomes than traditional statistical approaches [20,21]. Therefore, ML may enhance the understanding of PU predictors among healthcare providers in long-term-care settings. 

This study compared different ML methods, namely random forest, logistics regression, linear support vector machine (SVM), polynomial SVM, radial SVM, and sigmoid SVM, in terms of their accuracy, sensitivity, specificity, negative predictor values, and positive predictive values by validating real datasets in order to identify factors that affect PUs. Our prediction of PUs will help researchers and NH healthcare providers prevent PUs among NH residents.

## 2. Materials and Methods

### 2.1. Data Collection and Retrieval

The data used in this study were extracted from the original study (“Estimating Optimal Nurse Staffing for Nursing Home Residents Using an Optimization Model”), which was a longitudinal study where data were collected at seven time points over 3 years (May 2017–February 2020). Of these, only the dataset collected at the first time point (May–August 2017) was extracted and used. The data analyzed in this study were collected from 60 NHs across Korea. 

In the original study, organizational characteristics, NH staff characteristics, and residents’ characteristics were collected. Organizational characteristics of NHs included resident capacity, average number of current residents, long-term-care facility grade by Korean government, NH location, and ownership. NH staff (director, secretary general, social worker, dietician, administrative staff, registered nurses (RNs), certified nurse aides (CNAs), and care workers (CWs)) characteristics included number of staff, hours per resident day (HPRD), retention rate, and turnover rate. Lastly, NH resident characteristics included age, gender, and proportion of adverse outcomes (i.e., residents with cognitive dysfunction, urinary and/or fecal incontinence, aggressive behavior, depression, fall and/or slip prevalence, daily living help, hospital admission, decreased range of motion, weight loss, PU prevalence, dehydration, and/or those who are bed ridden, physically restrained, tube fed, and/or taking antidepressants or sleeping pills).

### 2.2. Data Preparation

For data preparation, variables unrelated to PUs were deleted, such as NH name, province, evaluation time (date of the long-term-care facility’s grade evaluation), and year of NH establishment. Among the independent variables, long-term-care facility grade is a nominal variable (Grade A is superior grade and Grade E is the lowest); thus, Grade A, B, C, D, E and Ungraded were scored as 5, 4, 3, 2, 1, 0, respectively.

### 2.3. Variable Selection

The variable PUs was selected as the dependent variable, which represents the number of NH residents who experienced PUs within 3 months of staying in the NH. In this study, there were a total of 57 independent variables related to Pus, comprising organizational factors (NH characteristics) and individual factors (NH resident characteristics). These factors were included after reviewing diverse previous studies about NH PUs. Moreover, the independent variables produced a large number of combinations (about 14,400,000 billion). Therefore, we selected 10 independent variables according to their importance score using random forest. Table 1 shows a list of significant variables sorted in order of importance score (3.0 and above) that are related to the occurrence of a PU. The threshold criterion was set to 3.0 based on discussions with several ML experts, as no threshold has been scientifically set.

### 2.4. Data Analysis

SPSS version 25.0 (IBM Corporation, Armonk, NY, USA) was used to analyze the collected data with means, standard deviations, frequency, and percentages. ML algorithms in R Studio v. 1.4.1106 (R Consortium, Boston, MA, USA) were used to develop a prediction model. We applied representative ML algorithms (random forest, logistic regression, linear SVM, polynomial SVM, radial SVM, and sigmoid SVM) in this study. The random forest algorithm is applicable when there are more predictors than observations and based on the theory of ensemble learning that allows the algorithm to accurately learn simple and complex classification functions. This algorithm does not require fine-tuning of parameters, and the default parameterization often leads to excellent performance [22]. The SVM model looks for a super-planes set in a high- or infinite-level space and uses them to perform classification and regression. SVM can efficiently learn complex classification functions and employs powerful regularization principles to avoid overfitting [23]. The linear SVM is a single-parameter classification function where the parameter biases the test along the SVM’s regression output values. The linear SVM is useful for handling large amounts of data vectors, such as text categorization. The polynomial SVM is often used to process images, and the radial SVM is a general-purpose method used when there is no prior information about the data. Sigmoid SVM is mainly used as a proxy for neural networks [24]. Five performance factors such as accuracy, sensitivity, specificity, PPV, and NPV measures were used to evaluate the models in this study.

### 2.5. Ethical Considerations

Ethical approval for this study was obtained from the institutional review board of Ewha Womans University in Korea (approval no. 136-4). Participating NHs understood the purpose and necessity of the research and agreed to participate. We ensured the protection of the participating NHs’ confidentiality.

## 3. Results

### 3.1. Participating NHs’ Organizational Characteristics

Table 2 shows the characteristics of participating NHs. The average number of current residents in the NHs was 70.03. A total of 38.3% of participating NHs were evaluated as a superior-grade NH, and 11.7% of NHs received a below-average grade by the Korean National Health Insurance Corporation. The HPRD for directors was 0.26 (15 min 36 s), 0.39 (23 min 24 s) for social workers, 0.09 (5 min 24 s) for dieticians, 0.14 (8 min 24 s) for administrative staff, 0.19 (11 min 24 s) for RNs, 0.36 (21 min 36 s) for CNAs, and 3.82 (3 h 49 min 12 s) for CWs. The retention rate was 78.71% for directors, 88.12% for secretary generals, 88.12% for social workers, 72.12% for dieticians, 68.72% for administrative staff, 81.72% for RNs, 76.82% for CNAs, and 67.72% for CWs. The average age of residents was 83.60, and the majority of them were female (78.95%). The percentage of bedridden NH residents was 25.91%, those prescribed antidepressants or sleeping pills were 26.73%, those with cognitive dysfunction and urinary incontinence were 61.56% and 41.10%, respectively, and those physically restrained were 7.40%.

### 3.2. Predictive Performance

Table 3 shows the predictive performance of the six models (random forest, logistic regression, linear SVM, polynomial SVM, radial SVM, and sigmoid SVM). Among them, the random forest model showed the greatest accuracy (0.843), followed by logistic regression (0.797), polynomial SVM (0.797), radial SVM (0.794), linear SVM (0.788), and sigmoid SVM (0.767). The random forest model showed the greatest sensitivity (0.513). After the sigmoid SVM (0.200), logistic regression (0.150), radial SVM (0.138), polynomial SVM (0.138), and linear SVM (0.125) had decreasing orders of performance. Moreover, the SVM polynomial model showed the greatest specificity (0.996), followed by linear SVM (0.992), radial SVM (0.989), logistic regression (0.977), random forest (0.955), and sigmoid SVM (0.943). The polynomial SVM model showed the greatest PPV (0.917). After the random forest model (0.846), logistic regression (0.769), radial SVM (0.732), linear SVM (0.727), and sigmoid SVM (0.500) had decreasing orders of performance. Finally, the random forest model showed the greatest NPV (0.865), followed by logistic regression (0.801), polynomial SVM (0.792) and radial SVM (0.792), linear (0.789), and sigmoid SVMs (0.767).

The strongest predictor of PUs in terms of staffing variables was director HPRD, whereas the strongest predicting variables related to residents’ characteristics were the proportion of residents who were bedridden, taking antidepressants or sleeping pills, and experiencing cognitive dysfunction.

Table 4 displays the optimal combination of variables with the highest accuracy. The table includes many variables associated with residents’ quality of care.

We developed a model with 10 features and performed an experiment. For example, each model can choose 10 features, respectively, and we can create 10 models. The model with 2 features can produce 45 models because it chooses 2 features not considering orders. Figure 1 shows the highest accuracy among the models. Random forest was the most accurate, and a model with 6 features offered the best function. Generally, the number of features does not correspond with the accuracy. 

## 4. Discussion

The residents’ health status were very strong predictors of PUs. Bedridden residents with less mobility were more likely to experience PUs, and the use of antidepressants or sleeping pills and cognitive dysfunction were also related to PUs. According to previous research, most residents with PUs had more bedridden days [25] than those without PUs the more active residents. This is because the bedridden state interrupts blood flow and damages residents’ skin [26]. In this respect, repositioning residents is the main PU preventative method [25]. Therefore, it is necessary to educate nursing staff who take care of residents on the importance of repositioning and supervise whether they are conducted repositions regularly. In the case of medication, even though few studies have investigated the effect of medications on PUs, one recent study revealed that corticosteroids could be a risk factor for PUs [27]. However, the relationship between PUs and antidepressants or sleeping pills is unclear. One possible explanation for this result is that these drugs induced less mobility, which might affect PUs. Because ML found that antidepressants or sleeping pills affect PUs, future studies should be conducted to obtain more evidence. In addition, previous studies with hip-fracture patients reported that the incidence of PUs increased in patients with severe cognitive dysfunction [28]. However, there is limited knowledge regarding the effect of cognitive function on NH residents’ PUs. One study found that NH residents who have severe cognitive impairment have problems describing their pain to their healthcare providers [29]. In a similar context, cognitive impairment might cause or worsen the development/prognosis of PUs because cognitive impairment hinders communication. Further studies on the effect of cognitive functions on PU development and care processes are needed. Based on the present evidence, NH healthcare providers should assess residents systematically about their mobility, use of medication, and cognitive function.

This study also highlighted the importance of NH directors. The results of this study are consistent with those of previous studies in that directors’ HPRD influenced residents’ health outcomes [12,30,31]. According to prior research, the director’s management philosophy and awareness of the importance of employees and finance as well as the manager’s experience and stability affected residents’ health outcomes [31]. In other words, a NH director’s leadership plays a decisive role in improving residents’ health outcomes. The director’s positive leadership can promote patient safety and a positive working environment and staff culture in NHs, which are important factors in achieving better resident health outcomes [32]. Especially, PU prevalence in NHs has been shown to be negatively associated with staff cohesion and the presence of self-managed teams [29]. Creating a cohesive culture among staff and forming a PU management team are crucial roles of the NH director [30,33]. A longitudinal study is needed to explore how the NH director’s role contributes to the decrease in the occurrence of Pus, and further research is also needed to examine the extent to which HPRD increases and the PU prevalence decreases.

Traditional statistical analyses report only limited findings because of problems with big data. However, ML has advanced enough to examine raw datasets without data sophistication or designated variables, and researchers are able to capture all the possible predictors that may not be found when using other statistical methods. This research applied diverse ML methods to find the best fit to identify the predicting factors of PUs. Future research should expand the scope of evaluation using different ML methods to investigate other possible factors of PUs.

## 5. Conclusions

This study applied six ML methods to predict the factors related to the prevalence of PUs in NH residents. The study was conducted based on data from 60 NHs distributed throughout Korea and their residents. The random forest model showed the greatest accuracy (0.814). Subsequently, the logistic regression (0.782), SVM polynomial (0.779), SVM radial (0.770), SVM linear (0.767), and SVM sigmoid (0.674) sequentially followed. Based on these findings, the random forest model is a powerful algorithm to predict PUs in NHs.

Many factors that predict PUs in NHs were identified according to diverse ML methods, including NH characteristics (directors’ HPRD, CNAs’ HPRD, number of current residents, superior grade in facility evaluation, and CWs’ retention rate) and resident characteristics (proportion of bedridden residents, those taking antidepressants or sleeping pills, and those experiencing cognitive dysfunction, urinary incontinence, and restraint). Therefore, both NH and residents’ characteristics should be considered to decrease PUs among NH residents.

## Figures and Tables

**Figure 1 ijerph-18-02954-f001:**
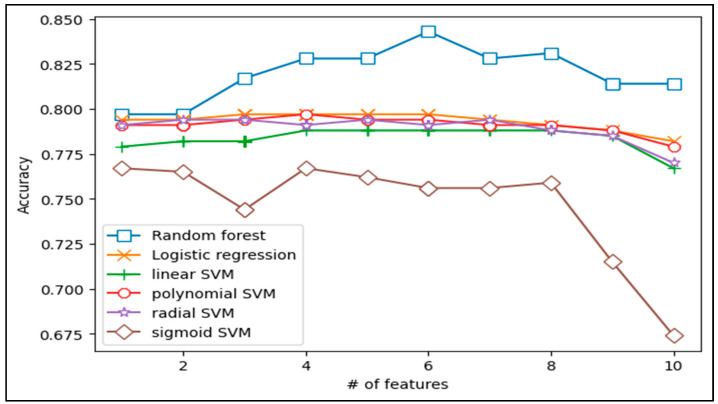
Comparison of Accuracy in Prediction Models.

**Table 1 ijerph-18-02954-t001:** Variables’ Importance Scores

No.	Variable	Importance Score
1	Hours per resident day of director	4.431
2	Proportion of bedridden residents	4.387
3	Proportion of residents taking antidepressants or sleeping pills	4.129
4	Proportion of residents with cognitive dysfunction	4.031
5	Proportion of residents with urinary incontinence	3.862
6	Proportion of residents with restraint	3.700
7	Hours per resident day of the certified nurse aide	3.411
8	Number of current residents	3.150
9	Ratio of Grade A	3.105
10	Retention rate of care worker	3.013

**Table 2 ijerph-18-02954-t002:** Characteristics of Nursing Homes

Variable	Label (Range)	*n*	%	M ± SD
Average number of current residents				70.03±51.11
Long-term care facility grade (%)	Grade A (Superior) ^a^	23	38.3	
	Grade B (Above average) ^b^	8	13.3	
	Grade C (Average) ^c^	6	10.0	
	Grade D (Below average) ^d^	7	11.7	
	Grade E (Poor) ^e^	16	26.7	

HPRD of staff	Director			0.26 ± 0.23
	Secretary general			0.12 ±0.12
	Social worker			0.39 ± 0.27
	Dietician			0.09 ± 0.09
	Administrative staff			0.14 ± 0.22
	Registered nurses			0.19 ± 0.24
	Certified nurse aides			0.36 ± 0.26
	Care worker			3.82 ± 1.63
Retention rate of staff	Director			78.71 ± 20.78
	Secretary general			79.72 ± 31.16
	Social worker			88.12 ± 20.21
	Dietician			72.12 ± 20.11
	Administrative staff			68.72 ± 23.19
	Registered nurses			81.72 ± 30.29
	Certified nurse aide			76.82 ± 28.29
	Care worker			67.72 ± 30.22
Age				83.60 ± 2.40
Gender (%)	Female			78.95 ± 11.30
	Male			20.77 ± 11.35
Quality of care of residents	Cognitive dysfunction		61.56	
	Urinary Incontinence		41.10	
	Antidepressants or sleeping pills		26.73	
	Fecal Incontinence		21.42	
	Bedridden		25.91	
	Physically restrained		7.40	
	Tube feeding		8.66	
	Aggressive behavior		4.62	
	Depression		5.55	
	Fall prevalence		4.84	
	Help for daily living		4.27	
	Slip prevalence		3.36	
	Hospital admission		2.69	
	Range of motion		2.52	
	10% Weight loss		1.68	
	5% Weight loss		1.12	
	Pressure sore prevalence		1.21	
	Dehydration		0.73	

*Note*. SD = standard deviation; ; HPRD = hours per resident day. ^a^ Score of 90 or more, and 70 points or more of each major classification area. ^b^ Score of 80 or more, and 60 points or more of each major classification area. ^c^ Score of 70 or more, and 50 points or more of each major classification area. ^d^ Score of 60 or more, and 40 points or more of each major classification area. ^e^ Score of 59 or less, and 39 points or less in each major classification area.

**Table 3 ijerph-18-02954-t003:** Comparison of Performance of Prediction Models.

Model	Accuracy	Sensitivity	Specificity	PPV	NPV
Random forest	0.843	0.513	0.943	0.732	0.865
Logistic regression	0.797	0.200	0.977	0.727	0.801
Linear SVM	0.788	0.125	0.989	0.769	0.789
Polynomial SVM	0.797	0.138	0.996	0.917	0.792
Radial SVM	0.794	0.138	0.992	0.846	0.792
Sigmoid SVM	0.767	0.150	0.955	0.500	0.767

*Note.* PPV = positive predictive values, NPV = negative predictive values, SVM = support vector machine.

**Table 4 ijerph-18-02954-t004:** Combination of Variables in Prediction Models

Model	Combined Variables
Random forest	Grade A + CAN HPRD + HPRD of director + urinary incontinence + medication + restraint
Logistic regression	Grade A + cognitive dysfunction + bedridden
Linear SVM	Average number of current residents + Grade A + urinary incontinence + bedridden
Polynomial SVM	Average number of current residents + cognitive dysfunction + urinary incontinence + restraint
Radial SVM	Average number of current residents + Grade A + HPRD of CNA + retention rate of CW + cognitive dysfunction
Sigmoid SVM	Grade A + HPRD of director + bed ridden + medication

*Note.* CNA = Certified nursing aide; CW = care worker; HPRD = hours per resident day

## Data Availability

Not available.

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
