# Peer review of "Identifying the Risk Factors Associated with Nursing Home Residents’ Pressure Ulcers Using Machine Learning Methods"

_ijerph, 2021, doi:10.3390/ijerph18062954_

Round 1
Reviewer 1 Report
I think the paper is very interesting and original providing evidence on how advance thecnology can be implemented to explore the problem of UP.
I would have just a few suggestions:
Lines 29-31. These lines are quite strong. I think they represents only a part of the reasons for NH population frailty (even if very important and often underestimated). Regarding sexual abuse one of the papers you cited evidence that: “None of the six selected articles can tell anything about the prevalence of sexual abuse of older residents in institutions in numbers”; and again: ”at this stage impossible to estimate the prevalence of sexual abuse of nursing home residents”. The other reference present results of a study conducted in Switzerland.
Maybe I would consider to reword a bit these lines because it cannot be taken for granted that these results can be generalized to the whole NH residents’ population.
Line 37. I would suggest to specify to which population these data refer to. Something like: “Furthermore, according to a study conducted in the United States on a population of almost 3 million of NH residents…..etc”
Lines 76-76. Has the original study been published already? Do you have any reference to add?
Lines 230-233: I imagine these results refer to Table 1 and results from the implementation of the "random forest" method? Is it correct? If so I would add this as an aim of the paper (you discussed these results even more than the comparison between different methods) and I suggest to underline that to give these results you used the method that showed the best accuracy data.
Lines 296-297: I could not open the link. It does not work (it gives me the following message: “Error 404 - File not found”).
Reviewer 2 Report
This study compared different ML methods in terms of the accuracy, sensitivity, specificity, negative predictor values, and positive predictive values by validating real datasets to predict factors for PUs.
My comments are as follows:
Method
- It is better for the authors to state why the reason that “The original study was a longitudinal study in which a total of seven data collections were conducted over 3 years (May 2017–February 2020)…” but only the first collected data (May 2017–August 2017) were extracted and used.
- In data analyses, the authors should give necessary information about data used, including
- How many observations were in each nursing home and in this study?
- What variables and how many were input in the data analyses model?
- In variable selections (page 3, table 1), you stated that significant variables sorted in order of importance scores (3.0 and above)? You need to describe the necessary references to support your variable selections. Why were the variables in terms of nursing homes (organizational) and residents’ characteristics were chosen? Or have you put all the variables related in random forest tree to run the importance scores?
- Since you give information about variable selections in table 1, it is confused that what variables were used in other ML methods? All the same variables were used or other methods in terms of variable selections in other ML methods?
- There was no grade E but 16 ungraded, you need also to explain how did you deal with the variable in ML.
Results
- HPRD?? you need to note this somewhere formally before showing the abbreviation.
- In Table 2. NH Characteristics, what you mean by age and gender? Who’s age and gender? Why did you not show also all variables in terms of residents’ characteristics (Proportion of residents with Bed ridden, Proportion of residents with Antidepressant or sleeping pill, Proportion of residents with Cognitive Dysfunction)??? Besides, the format of table 2 seems misplaced in the pdf file.
- You show only the information in terms of Accuracy, Sensitivity, Specificity, PPV, NPV in Table 3. There were no information regarding your classification process and results in the random forest tree. Besides, there was no results to support your conclusion that “The residents’ health statuses were very strong predictors of PUs....” other than table 1 for the variable selections. I suggest you need to give more precise information in the main text.
Discussions
- What do the authors mean by traditional statistical method? logistic regression? or some other method?
- It is necessary to be more specific showing what the difference and the disadvantage of outcomes between ML and traditional method in this case. Why the ML method of random forest tree is better?
References
- More than 50% references were before 2016.
- a “e” is missed in the title of 2.3 “Variable Selction”. Please check the typolog.
